# Plant Regeneration via Organogenesis in Jerusalem Artichokes and Comparative Analysis of Endogenous Hormones and Antioxidant Enzymes in Typical and Atypical Shoots

**DOI:** 10.3390/plants12223789

**Published:** 2023-11-07

**Authors:** Yiming Zhang, Jiahui Zhang, Junliang Yin, Yiqing Liu, Xiaodong Cai

**Affiliations:** 1Spice Crops Research Institute, College of Horticulture and Gardening, Yangtze University, Jingzhou 434025, China; zhangyiming025@foxmail.com (Y.Z.); Jiahui.zhang0901@foxmail.com (J.Z.); 2College of Agriculture, Yangtze University, Jingzhou 434025, China; w.yinzi@163.com

**Keywords:** atypical shoots, endogenous auxin, genetic stability analysis, *Helianthus tuberosus*, plant tissue culture

## Abstract

The Jerusalem artichoke (*Helianthus tuberosus*) is a tuberous plant with considerable nutrient and bioactive compounds. The optimization of the in vitro clonal propagation protocol is critical for large-scale reproduction and biotechnological applications of Jerusalem artichoke production. In this work, in vitro plant regeneration from the stem nodes of the Jerusalem artichoke via direct organogenesis is presented. In the shoot induction stage, the stem segments produced more shoots with vigorous growth on MS medium containing 0.5 mg/L 6-benzylaminopurine (6-BA). The concentrations of 6-BA and gibberellic acid (GA_3_) were both optimized at 0.5 mg/L for shoot multiplication, and the combination of 0.05 mg/L indole-3-butyric acid (IBA) and 0.05 mg/L 1-naphthylacetic acid (NAA) was the most responsive for root induction, yielding the largest number of roots. The regenerated plantlets were successfully hardened at a 96% survival rate and vigorously grew in the field. The genetic stability of the regenerated plants was confirmed by flow cytometry and simple sequence repeat (SSR) analysis. However, 17.3% of shoots on the optimum shoot induction medium had withered leaves and excessive callus (atypical shoots), which greatly reduced the induction efficiency. Enzyme activity in the typical and atypical shoots was compared. The atypical shoots had significantly higher levels of endogenous indole-3-acetic acid (IAA) and abscisic acid (ABA), as well as increased activity of catalase (CAT), peroxidase (POD), and superoxide dismutase (SOD), whereas the content of 6-BA, zeatin (ZT), and GA_3_ was significantly reduced. The activity of the three enzymes was positively correlated with the content of IAA and ABA, while being negatively correlated with that of 6-BA, ZT, and GA_3_. The results suggest that the poor growth of the atypical shoots might be closely related to the significant accumulation of endogenous IAA and ABA, thus significantly increasing antioxidant enzyme activity.

## 1. Introduction

The Jerusalem artichoke (*Helianthus tuberosus* L.) is a perennial tuberous herb crop species belonging to the Asteraceae family. It is native to eastern North America, from which it was gradually introduced into Europe in the seventeenth century and later spread to Asia, including China [1]. The crop is widely distributed in a wide range of ecoclimatic regions in China, such as boreal, montane, and coastal environments, owing to its high adaptability and strong stress resistance [2]. Its underground tubers contain considerable amounts of carbohydrates and abundant fructose polymers, such as inulin and starch, and are often consumed as a vegetable and used for functional food applications, pharmaceutical purposes, and biofuel production [3,4,5]. Furthermore, its aerial parts are characterized by a rapid growth rate. They are rich in nutrients and bioactive substances and have a satisfactory digestion performance, therefore, they can serve as a neotype feed resource [3]. In addition to its agricultural, medicinal, and industrial benefits, this species is also cultivated for ornamental and ecological purposes in China [2].

The propagation of large quantities of high-quality propagules is required for successful crop production in the field. The Jerusalem artichoke shows a strong self-incompatibility response, resulting in very low self-fertility and poor seed production, and the seeds are usually difficult to germinate [6]. This makes reproduction via seeds a non-viable propagation approach. At present, the Jerusalem artichoke is mainly vegetatively propagated by tubers [7]. Tubers can be infected by plant pathogens, such as *Sclerotinia sclerotiorum* and *Sclerotium rolfsii*, during cultivation and tuber storage, which may lead to stem and/or tuber rot and thus cause substantial yield losses and a large reduction in tuber quality [8,9]. These diseases can be transmitted through the infected mother tubers after continuous intensive cultivation. As a consequence, the traditional tuber propagation may pose serious threats to the reproduction of Jerusalem artichokes. Plant tissue culture has been regarded as one of the most fundamental and efficient tools in the agricultural industry. It is not only routinely employed in clonal propagation, germplasm conservation, and secondary metabolite production in plants, but has also emerged as an essential component of plant biotechnology and genetic engineering [10,11,12]. Therefore, the establishment of an efficient and stable in vitro regeneration system is necessary for large-scale multiplication and genetic improvement in the case of Jerusalem artichokes.

During the past few decades, in vitro culture techniques have been explored for Jerusalem artichokes for rapid clonal propagation [13,14], microtuber induction [15], germplasm cryopreservation [16], inulin production [15], seed dormancy breaking [6], and genetic transformation [17]. Generally, callus induction is an important step in the in vitro regeneration protocols of many plants [18,19,20]. In the case of Jerusalem artichokes, researchers have shown that calli could be readily induced from tubers, leaves, and stem cuttings on MS basal medium supplemented with different types of plant growth regulators (PGRs) used in different concentrations [14,17,21,22]. Yan et al. [22] found that the induced callus rapidly grew with a strong proliferation ability, but the authors had difficulties in obtaining differentiation into shoots or somatic embryos. A contradictory result was reported by Taha et al. [21], who stated that shoots could be regenerated from calli induced from the nodal explants and leaves of Jerusalem artichokes via organogenesis. Plant regeneration through direct somatic embryogenesis was also reported in that particular crop, where globular somatic embryos were initiated on the leaf adaxial surface without an intermediate callus phase [13]. In addition, both organogenesis and direct somatic embryogenesis were observed when culturing the leaf segments of Jerusalem artichokes in the presence of ZT [17]. Despite some successful cases in plant regeneration via callus or somatic embryogenesis at present, the rapid clonal production of Jerusalem artichokes is mainly conducted through direct organogenesis with nodal stems as explants [7,14,22]. However, there remain a few challenges to be addressed during the shoot and root induction of Jerusalem artichokes, including large and dense calli formed at the explant base, a high percentage of hyperhydricity (vitrification), and withered leaves [17,22]. 

Plant endogenous hormones play an important role in almost all processes of plant growth, development, senescence, and stress responses [23,24]. During plant tissue culture, they are considered to be one of the crucial factors regulating plant organogenesis and morphogenesis [25,26]. Auxins, cytokinins, gibberellins, abscisic acid, and ethylene are commonly considered to be the main natural plant hormones. Recently, the effects of the endogenous hormone content on in vitro culture responses have been studied in a variety of plant species, such as *Tulbaghia simmleri* [27], *Pinus koraiensis* [28], and *Vitis vinifera* [29]. The morphogenetic abilities are different in different explant types, which may be attributed to the differences in their endogenous hormone content [28,30]. Under in vitro culture conditions, the explant differentiation, plant regeneration, and physiological processes are modulated by the interaction of the PGRs and endogenous hormones [23,27]. By adding different exogenous PGRs to the culture medium, distinct changes both in the endogenous hormone levels and physiological and biochemical reactions were observed in the cultured products [27,31]. Antioxidant enzymes can scavenge reactive oxygen species and lipid peroxidation products, and their activity is commonly enhanced when suffering from biotic or abiotic stresses [32]. Like excess PGRs, when under specific in vitro culture conditions, the in vitro cultures may suffer from oxidative stress and, consequently, will regulate the endogenous hormone content and antioxidant enzyme activity to alleviate oxidative damage [33,34]. Schnablova et al. [35] reported that the overproduction of endogenous cytokinins increased the activity of some antioxidant enzymes involved in lignification, such as syringaldazine peroxidase and guaiacol peroxidase. These previous findings provide foundations for further optimization of the exogenous PGRs in the culture medium, with the aim to establish an ideal in vitro culture protocol. However, no research has been carried out to investigate the underlying physiological mechanisms during the in vitro morphogenesis of Jerusalem artichokes.

The purpose of the current investigation was to first develop a rapid and stable in vitro plant regeneration protocol for the Jerusalem artichoke via organogenesis by using different concentrations of exogenous PGRs, including 6-BA and gibberellic acid (GA_3_) for shoot induction and multiplication, as well as indole-3-butyric acid (IBA) and 1-naphthylacetic acid (NAA) for root induction. Furthermore, the content of endogenous hormones and the antioxidant enzyme activity in the regenerated shoots differing in phenotypic characteristics were evaluated with the aim of understanding the possible physiological mechanisms involved in the formation of atypical shoots in the Jerusalem artichoke.

## 2. Results

### 2.1. Surface Disinfection with HgCl_2_ Solution

The effects of different treatment times with 0.1% (*w*/*v*) HgCl_2_ solution on the contamination rate and survival rate of the stem cuttings are shown in Figure 1. The 6 min treatment was found to be the most efficient for surface disinfection with a 0% contamination rate in comparison to the other three different exposure times, whereas it was adverse to the sprouting of the lateral buds and significantly reduced the survival rate (80.7%) compared with the disinfection time of 2 (88.0%) and 4 min (85.9%). The 4 min treatment resulted in a significantly lower contamination rate (3.7%) than the treatments of 1 (20.3%) and 2 min (11.5%), and the survival rate was similar to that of the 2 min treatment, and was significantly higher than that of the 1 min treatment (79.7%). It might be suggested on the basis of the obtained results that incubation in 0.1% HgCl_2_ for 4 min is the optimum for the surface disinfection of the stem nodes of Jerusalem artichoke.

### 2.2. Effects of Different Concentrations of 6-BA on Shoot Induction 

Four shoot induction (SI) media, namely MS basal medium supplemented with 0 (SI1), 0.25 (SI2), 0.5 (SI3), and 1.0 mg/L 6-BA (SI4), were used for shoot induction. After 6 days of culture, shoot initiation was observed on the four media, including the SI1 medium without 6-BA. Thirty days later, the shoots regenerated on the SI1 medium had a longer shoot length and larger leaves (Figure 2A) compared with the other three media, whereas both the SI2 (Figure 2B) and the SI4 medium (Figure 2D) appeared to have an inhibitory effect on the development of the regenerated shoots and leaves. Different amounts of calli were also formed spontaneously at the explant base on all four applied media (Figure 2A–D), and large-sized callus formation was observed on the SI3 (Figure 2C) and SI4 (Figure 2D) media containing 0.5 and 1.0 mg/L 6-BA, respectively. Furthermore, some of the shoots were observed to have an atypical phenotype on all the media containing 6-BA. On the SI3 medium, although the majority of the regenerated shoots exhibited a normal morphology (typical shoots), a few shoots characterized by etiolated and even withered leaves, retarded elongation, and excessive calli (atypical shoots) were also observed (Figure 2E). 

As shown in Table 1, most of the explants developed into shoots with significantly different phenotypic characteristics depending on the concentration of 6-BA added in the culture medium. With the increase in the 6-BA concentration, the shoot induction frequency declined, whereas the number of regenerated shoots per explant increased. The lowest shoot induction frequency (85.2%) was observed on the SI4 medium, followed by the SI3 medium (87.2%), and both of which were significantly lower than on the SI1 medium (94.4). On the SI4 medium, the number of regenerated shoots per explant reached 3.6, which was significantly larger as compared to the other media. Especially on the SI1 and SI2 media, the shoot number was only 1.1 and 1.5 per explant, respectively. Shoot quality was determined by the node number per shoot and shoot length in this study, and the results showed that both of them initially decreased significantly, then increased significantly, and finally reduced significantly again with the increase in 6-BA. The largest node number per shoot was observed on the SI3 medium (6.5), which was approximately 1.7 and 1.5 times larger than those cultured on the SI2 and SI4 media. The shoots produced on the SI4 medium had the shortest shoot length (1.8 cm), which was significantly shorter as compared to the other treatments, except for the shoots produced on the SI2 medium. Concerning the fresh weight of the callus formed at the cutting end, an obvious increasing trend was noted with the increase in 6-BA. In addition, the frequency of atypical shoots was significantly increased with the increase in the 6-BA concentration. The highest frequency of atypical shoots was observed on the SI4 medium (28.0%), followed by the SI3 (17.3%) and the SI2 media (8.3%), and no atypical shoots were observed on the SI1 medium.

### 2.3. Effects of Different Concentrations of 6-BA and GA_3_ on Shoot Multiplication

After 30 days of culture on the shoot multiplication (SM) media containing different concentrations of 6-BA and GA_3_, the regenerated multiple shoots demonstrated large differences in the shoot number per explant, shoot length, and leaf number per shoot (Table 2). The maximum shoot number per explant (3.7) was found on media containing 1.0 mg/L 6-BA and 0.5 mg/L GA_3_ (the SM4), which was significantly higher than those cultured on the media containing a lower concentration of GA_3_. However, a higher concentration of 6-BA and GA_3_ (the SM4) was not conducive to shoot growth and development, and it significantly reduced the shoot length (1.7 cm) and leaf number (4.3) compared with the others. In the cases of 0.5 mg/L 6-BA and 0.5 mg/L GA_3_ (the SM2 medium), the regenerated shoots were significantly longer (3.3 cm) and had a significantly higher leaf number (6.1 per shoot) than those cultured on the SM3 and SM4 media. No significant differences in the measured values were observed between the SM1 and SM2 treatments, both of which had a lower concentration of 6-BA. Two-way ANOVA suggested that the shoot number per explant was significantly dependent (*p* < 0.05) on the GA_3_ concentration in the culture medium. The 6-BA concentration had a significant impact on the leaf number per shoot (*p* < 0.001). The concentrations of both 6-BA (*p* < 0.001) and GA_3_ (*p* < 0.01) and their interaction (*p* < 0.001) significantly affected the shoot length. Furthermore, a larger and more compact callus was observed at the base of the stem nodes on the SM3 and SM4 media, which contained a higher concentration of 6-BA (Figure 3). On the SM1 (Figure 3A) and SM2 media (Figure 3B), the regenerated multiple shoots grew normally with healthy leaves. However, some of the regenerated shoots on the SM3 (Figure 3C) and SM4 media (Figure 3D) were observed to be withered or even dead. 

### 2.4. In Vitro Rooting and Plant Establishment in the Field

In this experiment, in vitro-multiplied shoots were excised (about 1.5 cm in length) and cultured on the four root induction (RI) media. Roots were initiated after about 7 days of culture, and several 3–5 cm roots were observed 12 days after culture on the RI2 medium containing 0.05 mg/L IBA and 0.05 mg/L NAA (Figure 4A). Thirty days later, whole plantlets (Figure 4B) with well-developed roots (Figure 4C) were regenerated on all the culture medium combinations. As presented in Table 3, the root induction frequency varied from 85.0% (the RI3 medium without NAA) to 95.8% (the RI2 medium containing 0.05 mg/L NAA), with no significant differences among the adopted four RI media. When cultured on the RI2 medium, both the adventitious root number per plantlet (4.7) and the average root length (14.7 cm) were the highest. The shoots cultured on the RI2 medium produced a significantly larger number of roots as compared with the other media, except for the RI4 containing 0.05 mg/L NAA, whereas the average root length did not differ significantly from those cultured on the other media. In addition, the root number observed on both the RI2 and RI4 media was significantly higher than that of the RI1 medium. According to the results of the two-way ANOVA, the NAA supplemented in the RI media had a significant influence on the adventitious root number per plantlet (*p* < 0.01). Furthermore, the two factors and their interaction had no significant effects on the root induction frequency, adventitious root number, and root length.

For plant establishment, in total, 300 well-developed plants regenerated from the stem nodes were acclimated ex vitro, and 288 plants (96% survival rate) survived on the substrates 30 days later (Figure 4D). Finally, 120 plants were successfully planted in the field, with a survival frequency of 100%. All these plants grew normally and vigorously with a similar phenotype after two months of planting in the soil (Figure 4E).

### 2.5. Genetic Stability Analysis via Flow Cytometry and SSR Markers

Flow cytometry and SSR marker analysis were used to evaluate the genetic stability of the tissue culture-derived plants. The results of the flow cytometry analysis of the donor plants and the in vitro regenerated plantlets are shown in Figure 5. Both the tuber-forced plants (Figure 5A) and all the analyzed micropropagated plants (Figure 5B) had only one main peak situated at a value of approximately 7 × 10^5^ in the histogram. Of the ten SSR primer combinations used (Appendix A), six pairs, including JS5, JS8, JS18, JS19, and JS20, produced clear and reproducible DNA bands in 8% polyacrylamide gels. In total, 247 alleles with the desired size at 19 loci were amplified by the six pairs of SSR primers (Figure 6A–F), including 52 alleles at four loci by JS5 (Figure 6A), 65 alleles at five loci by JS8 (Figure 6B), 13 alleles at one locus by JS13 (Figure 6C), 52 alleles at four loci by JS18 (Figure 6D), 26 alleles at two loci by JS19 (Figure 6E), and 39 alleles at three loci by JS20 (Figure 6F). No banding pattern changes were detected by the six pairs of SSR primers between the donor plants and the in vitro-derived group.

### 2.6. Comparative Analysis of the Endogenou Hormone Levels and Antioxidant Enzyme Activity in the Two Types of Shoots 

During in vitro culture, endogenous hormones are considered to play important roles in regulating plant organogenesis and morphogenesis [25,26]. As mentioned above, two types of shoots were observed on the SI3 medium, namely typical and atypical shoots (Figure 2E). Therefore, the two types of shoots were subjected to a comparative analysis of the endogenous hormone content and antioxidant enzyme activity to understand the physiological reasons underlying the striking morphological differences in these shoots. In total, one auxin (IAA), two cytokinins (6-BA and ZT), gibberellin (GA_3_), and ABA were detected as presented in Figure 7. Significant differences in the content of all the analyzed endogenous hormones were found between the two types of shoots (Figure 7A–E). As compared to the typical shoots, the atypical shoots accumulated (*p* < 0.01) a significantly larger amount of endogenous IAA (Figure 7A) and ABA (Figure 7E), whereas the content of 6-BA (Figure 7B), ZT (Figure 7C), and GA_3_ (Figure 7D) was significantly lower (*p* < 0.05 for 6-BA and ZT; *p* < 0.01 for GA_3_). An approximately 9-fold and 5.8-fold difference was detected in the GA_3_ and ABA content between the two types of shoots, respectively. 

Significant differences in the activity of the measured three antioxidant enzymes were also observed between the two types of shoots (Figure 8A–C). Results showed that the atypical shoots presented significantly higher CAT activity (Figure 8A), POD activity (Figure 8B), and SOD activity (Figure 8C) than the typical shoots (*p* < 0.05 for CAT; *p* < 0.001 for POD and SOD). 

### 2.7. Correlation Analysis 

Correlations between the endogenous hormone levels and antioxidant enzyme activity in the two types of shoots were analyzed by the Pearson correlation coefficient method (Figure 9). It was found that the activity of the three antioxidant enzymes displayed a positive correlation, and the CAT activity was significantly correlated with the SOD activity (r = 0.83). Among the content of the five analyzed hormones, no significant correlations were found except for an extremely significantly negative correlation between IAA and ZT (r = −0.94). The activity of all three enzymes showed a positive correlation with the content of both IAA and ABA, while it had a negative correlation with the content of 6-BA, ZT, and GA_3_. The POD activity displayed a significantly negative correlation with the levels of both 6-BA (r = −0.94) and GA_3_ (r = −0.89). A significantly negative correlation was also observed between the ZT content and the activity of both CAT (r = −0.83) and SOD (r = −0.89).

## 3. Discussion

In this study, the in vitro clonal propagation of Jerusalem artichoke was achieved via organogenesis by using the stem nodes as the initial material. Large and compact callus tissues were observed when the stem nodes were cultured on the medium containing 0.5 or 1.0 mg/L 6-BA), which might have an inhibitory effect on the axillary bud formation and shoot elongation. Therefore, we carried out a comparative analysis of the endogenous hormone levels and antioxidant enzyme activity in the typical and atypical shoots. This allowed us to study the physiological differences in shoot morphology, which could aid in establishing a more efficient in vitro protocol for the massive propagation of Jerusalem artichoke.

### 3.1. Establishment of In Vitro Propagation System

In vitro shoot induction and multiplication is a common procedure in plant tissue culture [23]. Precise concentrations and optimal ratios of plant growth regulators (PGRs) are essential in efficiently obtaining enough shoots of high quality in vitro [19,36]. It has been widely reported that 6-BA is the most frequently used cytokinin in tissue culture, and it is a determinant in promoting cell division and organ differentiation [37]. In the Jerusalem artichoke, 6-BA-containing media are often used to induce shoot emergence and development [14,17,22]. A result consistent with this was also obtained in this study, and the medium containing 0.5 mg/L 6-BA was proven to be the most efficient for the induction of shoots with vigorous growth. In addition, shoot proliferation is largely performed by shoot segments cultured on media containing cytokinins, such as 6-BA [37,38], and exogenously applied GA_3_ is reported to be efficient in stimulating shoot elongation [39]. In this study, 0.5 mg/L 6-BA in combination with 0.5 mg/L GA_3_ was found to be the most responsive for shoot proliferation and elongation. In previous reports, a certain concentration of GA_3_ was also successfully applied for shoot multiplication in the Jerusalem artichoke by Zhang et al. [16] and Kim et al. [17]. 

In addition, plant regeneration via organogenesis or somatic embryogenesis from the callus has been widely reported, which provides an alternative to propagating elite plant species [18,19,20]. The callus is amenable to induction from various explant types in the Jerusalem artichoke, such as the tubers, leaves, and stem cuttings [14,17,21]. In some previous studies, the induced callus was observed to have the capability of differentiation into shoots via organogenesis [17,22] or somatic embryogenesis [22]. However, in this study, the callus formed at the explant base had no shoot regeneration potential after 60 days of culture on the MS basal media supplemented with 0.5 or 1.0 mg/L BA and 0.3 or 0.6 mg/L NAA. Further study is needed to establish an ideal protocol for shoot induction from the callus by optimizing the PGR combinations and concentrations.

In plant tissue culture, shoot induction and multiplication usually come with several problems when extreme dosages of cytokinins or an unbalanced ratio of auxins to cytokinins are used, such as the production of excessive calli, retarded enlargement, and shoot tip necrosis [40]. Therefore, it is critical to seek a balanced concentration of PGRs that may reduce the callus size formed at the explant base and may not affect the efficiency of shoot induction and proliferation. In this study, large and compact calli were observed at the basal end of the stem nodes during shoot induction and multiplication, results showed that the higher the concentration of 6-BA in the medium, the larger the callus formed. Moreover, shoot development was observed to be inhibited by the massive callus formed at the explant base, thus resulting in a poor growth response in the induced shoots. In the Jerusalem artichoke, researchers also found that a large and compact callus, etiolated shoots, hyperhydricity, and even withered leaves occurred frequently when applying a relatively high concentration of 6-BA during shoot induction [17,22]. The phenomenon of massive calli formed at the explant base was also reported in several other plant species, such as *Trichosanthes kirilowii* [36], *Metasequoia glyptostroboides* [41], and *Lycopersicon esculentum* [42]. Most of these reports also found that the formed callus hindered the regeneration, elongation, and rooting of shoots [36,41].

### 3.2. Genetic Stability Assessment of In Vitro-Regenerated Plants

It has been extensively documented that somaclonal variation occurs amongst regenerants in most economically important plant species during tissue culture, with changes in phenotypic traits, ploidy levels, anatomical structures, biochemical characteristics, DNA sequences, or epigenetic levels [43,44,45]. Although somaclonal variation contributes to the creation of novel genetic variation that is useful for plant genetic improvement, it is undesirable when the aim of an in vitro culture system is clonal propagation for commercial production [43,45]. Therefore, an assessment of the genetic stability of the in vitro-regenerated plants is essential for a tissue culture procedure. Generally, the genetic fidelity of the regenerants can be confirmed by morphological, cytological, and DNA-based marker analysis [45,46]. In this study, the genetic homogeneity of the in vitro-produced plants was confirmed by flow cytometry and SSR analysis. No genetic changes in the in vitro-regenerated plants of the Jerusalem artichoke were reported by Zhang et al. [16] via SSR markers. Contrarily, when using random amplified polymorphic DNA (RAPD), inter-simple sequence repeat (ISSR), and start codon targeted (SCoT) marker techniques, polymorphisms were detected among in vitro-induced cultures of Jerusalem artichoke by Abdalla et al. [14]. Ioannidis et al. [47] found that SSR markers could not effectively detect all genetic variability in the micropropagated plants of *Cannabis sativa* because of its large genome size. Therefore, more molecular markers and more primers should be adopted to obtain more molecular information about regenerated plants in the future. In addition, the number of the analyzed plants was rather small in this study, and it may not have provided complete information on the genetic stability of the regenerated population.

### 3.3. Differences in the Endogenous Hormone Content and Antioxidant Enzyme Activity in the Typical and Atypical Shoots

Under in vitro culture conditions, plant hormones and their interactions with exogenous PGRs not only have a determinative effect on explant dedifferentiation and differentiation [23,24], but also can regulate the physiological status and phenotypical characteristics of the cultures [27,29]. During in vitro culture, the aberrant or unexpected phenotypes of the cultured products markedly reduce the regeneration efficiency [26,48]. In this investigation, although the SI3 medium containing 0.5 mg/L 6-BA was proven to be the most suitable medium for shoot induction and development, a number of atypical shoots were observed. To understand the physiological reasons underlying the striking morphological differences between the two types of shoots treated with the same concentration of exogenous 6-BA, the endogenous hormone content was compared. Based on the obtained results, it was shown that the atypical shoots had significantly higher content of IAA and ABA as well as significantly lower content of 6-BA, ZT, and GA_3_, suggesting that the morphological differences between the two types of shoots might be related to the differences in their accumulation of endogenous hormones. The results are in agreement with the results reported by Liang et al. [29], who found that considerable differences in endogenous hormone levels were measured between normal and abnormal types of embryos in grapevine (*Vitis vinifera*). Similar results were also obtained in *Pinus koraien*, and substantial changes in endogenous hormones were found among the different types of calli [28]. 

In this present study, the atypical shoots were characterized by large and compact calli formed at the basal end of the segment. Commonly, applications of a certain concentration and ratio of exogenous auxins to cytokinins can induce callus formation in vitro [23]. Nevertheless, in this study, two types of shoots were obtained from the stem nodes of Jerusalem artichoke cultured on the same medium containing 0.5 mg/L 6-BA. The influences of the types, concentrations, and combinations of endogenous hormones on callus induction and callus morphogenesis have been documented in various plants [49,50,51]. Endogenous IAA is a crucial factor determining callus induction [51,52], whereas endogenous ABA is usually considered an inhibitor and is involved in various environmental stress responses [23,53]. Endogenous cytokinins, including ZT and 6-BA, have multiple roles in plant tissue culture, such as the induction of callus and adventitious shoots, and GA_3_ can promote the elongation of the regenerated shoots [14,38,39]. In this study, the significantly increased production of IAA and ABA and significantly reduced levels of 6-BA, ZT, and GA3 were detected in the atypical shoots. These significant differences in endogenous hormone content between the typical and atypical shoots might be responsible for their obvious differences in growth and morphology. Since both types of shoots developed under the same conditions, the factors leading to the changes in endogenous hormone content needed to be investigated. It is reported that endogenous hormone content differs significantly among tenotypes and explants and thus has a great influence on callus formation [28,54,55]. For example, Mostafa et al. [55] found that the stem tip was the most suitable for callus induction among different parts of garlic (*Allium sativum*), and the analyzed endogenous hormone content in the stem tip was significantly higher than that in other explants. In this study, although node stems of a similar length were used as the initial material for shoot induction, the node segments were excised from different parts of the shoots. Therefore, the significant differences in the analyzed hormone content might also have been caused by the different node segments used for shoot induction.

Antioxidant enzymes, such as SOD, CAT, and POD, are known to cooperatively protect cell structures against oxidative stress, and their activity is generally increased when undergoing biotic or abiotic stresses [32,56]. Under unique conditions, such as excess PGRs, high NaCl, or mineral deficiencies, in vitro cultures suffer from damage caused by increased reactive oxygen species (ROS) production. Consequently, defense systems, including enzymatic antioxidant enzymes, are enhanced to alleviate the oxidative stress [57]. In some cases, phenotypic aberration, such as habituation or hyperhydricity, occur frequently during in vitro culture [26,48], and oxidative stress damage is thought to be one of the major causes [48,58]. In this study, significantly improved activity of CAT, POD, and SOD was also measured in the atypical shoots. We speculate that oxidative stress was strongly induced during shoot induction, which stimulated the defense mechanisms and thus increased the activity of the three antioxidant enzymes in the atypical shoots.

It has been reported that there is a direct or indirect correlation between different endogenous hormones and antioxidant enzymes, as well as other physiological substances, which collectively affects differentiation and morphogenesis during tissue culture [33,34]. In this study, the correlation analysis showed that the activity of all three enzymes was positively correlated with the content of IAA and ABA while negatively correlated with the content of 6-BA, ZT, and GA_3_. As mentioned, significantly higher content of IAA and ABA, as well as significantly lower content of 6-BA, ZT, and GA_3_ was measured in the atypical shoot samples. The results indicate that the poor growth of the atypical shoots might have been caused by the significantly increased content of IAA and ABA, thus significantly increasing the activity of CAT, SOD, and POD. 

## 4. Materials and Methods 

### 4.1. Plant Material 

Fresh tubers of a local variety of Jerusalem artichoke (*Helianthus tuberosus* L.), widely cultivated in Jingzhou, Hubei, China, were used in this study. The tubers were harvested in mid-October of 2021 and then incubated in plastic pots (26.5 cm height and 18 cm diameter) filled with sterilized sand for 20 days. For sprouting, the tubers were maintained in an artificial climate incubator at 25 ± 2 °C under 85% relative humidity in the dark, and sterile water was applied every three days to keep the sand moist. 

### 4.2. Explant Disinfection 

When the tuber-forced plants had grown up to 10 cm in height, young, sprouted shoots were collected, and all leaves were removed. After the stems were washed thoroughly under running tap water for 30 min, they were surface disinfected via immersion in 70% (*v*/*v*) ethyl alcohol (Sinopharm Chemical Reagent Co., Ltd., Shanghai, China) for 30 s and then agitated in 0.1% (*w*/*v*) mercuric chloride (HgCl_2_) with two drops of Tween 20 for 1, 2, 4, and 6 min, respectively. Next, the stems were rinsed five times in sterile water and were cut into nodal sections of about 1.5 cm in length and placed on Murashige and Skoog (MS) basal medium [59] supplemented with 30 g/L sucrose and 7.3 g/L agar (pH 5.8). Two pieces of explants were incubated in each 100 mL Erlenmeyer flask containing 30 mL culture medium. Each disinfection treatment consisted of three replicates, and 12 explants were used in each replicate. After 30 days of culture at 25 ± 1 °C and fluorescent light of 45 μmol m^−2^ s^−1^ for 14 h photoperiod, the contamination rate and survival rate of the explants were estimated. The survival rate was scored as the percentage of the explants that produced shoots and remained contaminant-free. 

### 4.3. Shoot Induction and Multiplication

The obtained stems in the initiation stage were used as explants. For shoot induction (SI), the stem nodes were inoculated in MS basal medium supplemented with 0, 0.25, 0.5, or 1.0 mg/L 6-BA. Thirty days later, the shoot induction frequency (the percentage of explants producing shoots), number of regenerated shoots per explant, node number per shoot, shoot length, and frequency of atypical shoots (the percentage of shoots with etiolated leaves and excessive calli) were counted, and the fresh weight of the calli formed at the base was estimated. For shoot multiplication (SM), stem nodes (about 1.5 cm) were excised from the regenerated shoots and transferred to MS basal medium containing 0.5 or 1.0 mg/L 6-BA in combination with gibberellic acid (GA_3_) at a concentration of 0.1 or 0.5 mg/L. All the SI media and SM media contained 30 g/L sucrose and 7.3 g/L agar at pH 5.8. Three explants were inoculated in each 300 mL glass container with a plastic lid containing 60 mL medium. Twelve stem nodes were used in each treatment with three replications, and all cultures were incubated, as mentioned above, for 30 days. 

### 4.4. Induction of Rooting

The regenerated multiple shoots were excised and separated from the explants, and an individual shoot (about 1.5 cm in length) was placed on the root induction (RI) medium containing four combinations of indole-3-butyric acid (IBA; 0.05 and 0.1 mg/L) and 1-naphthylacetic acid (NAA; 0 or 0.05 mg/L) for whole plant regeneration. Each medium consisted of 1/2 MS salts and vitamins, 20 g/L sucrose, 7.3 g/L agar, and 0.5 g/L activated charcoal (AC), and the pH was set at 5.8. Three shoots were cultured in each 300 mL glass vessel filled with 60 mL medium. Three replicates with 12 explants each were cultured under the conditions described above. After 30 days of culture, the root induction frequency, adventitious root number per plantlet, and root length were estimated.

### 4.5. Acclimatization and Transplantation

Thirty-day-old plants with well-developed roots were transplanted individually to transparent plastic cups (top diameter: 9.0 cm, bottom diameter: 9.0 cm, and height: 15.0 cm) filled with sterilized culture substrates after the medium residues were fully washed off from the roots. The substrates were composed of peat moss (pH 5.6) and perlite at a 1:1 (*v*/*v*) ratio. To maintain relatively high humidity during acclimatization, each transplanted plant was covered with a plastic film. The plants were acclimated inside a growth chamber with an ambient temperature of about 25 °C at a 14 h light photoperiod with a light intensity of about 150 μmol m^−2^ s^−1^ and 60–70% relative humidity. About 15 days later, the plastic films were removed, and the plants were transferred into a greenhouse with temperatures ranging from 18 °C (night) to 32 °C (day). After acclimation for 25 days, the hardened plants were planted in the field. 

### 4.6. Flow Cytometry Analysis 

Twelve plants randomly selected from the regenerated plants were subjected to flow cytometry, with the tuber-propagated plants as a control. Flow cytometry analysis was conducted to measure the relative nuclear DNA content on a CytoFLEX flow cytometer (Beckman Coulter, Suzhou, China) with a 488 nm/638 nm laser configuration. The samples were processed according to the protocol of Zhang et al. [60] with minor modifications. Fresh tender leaves were cut into pieces of 1.5 × 1.5 cm^2^ after removing the midrib, chopped in 0.5 mL of ice-cold Tris-MgCl_2_ buffer [61] and 0.1 mL of 1 mg/mL RNase A, and then filtered through a 40 μm mesh sieve. Thereafter, 50 μL of 100 μg/mL propidium iodide (PI) was added to the nuclear suspension. The suspension was shaken gently to mix well and maintained in the dark for 3 min before sample analysis. The measurement of each sample was repeated three times.

### 4.7. SSR Analysis

Total genomic DNA was extracted and purified from the fresh leaves according to the method of Fulton et al. [62]. For all plant samples, polymerase chain reactions (PCR) were performed using 10 randomly selected pairs of simple sequence repeat (SSR) primers (Appendix A) retrieved from Yang et al. [63]. Each 20 μL of reaction mix was composed of 2 μL temple DNA (25 ng/μL), 1.0 μL of each forward and reverse primer (10 μM), 10 μL of 3G Master Mix for PAGE (Vazyme, Nanjing, China), and 6.0 μL ddH_2_O. The reactions were conducted in an FC-96G thermocycler (BigFish, Hangzhou, China) as follows: initial denaturalization of the DNA at 94 °C for 2 min, followed by 40 cycles of amplification consisting of denaturation at 94 °C for 30 s, annealing at a temperature depending on the primer pairs for 30 s, and extension at 72 °C for 30 s, with a final extension at 72 °C for 10 min. PCR products were analyzed by electrophoresis in 8% polyacrylamide gels under non-denaturing conditions according to Yu et al. [46].

### 4.8. Analysis of Endogenous Hormone Content and Antioxidant Enzyme Activity 

The typical and atypical shoots originated from nodal explants cultured on the SI3 medium were used for the analysis of the endogenous hormone content and antioxidant enzyme activity. Collected samples were surface-dried and then immediately frozen in liquid nitrogen and stored at −80 °C. Five endogenous hormones, namely 6-BA, zeatin (ZT), indole-3-acetic acid (IAA), abscisic acid (ABA), and GA_3_, were determined by high-performance liquid chromatography (HPLC). Hormones were extracted and purified according to the protocol of Ge et al. [64], with some modifications. Briefly, the frozen stems were ground into powders in liquid nitrogen, and then 0.5 g samples were homogenized in 1 mL cold 80% (*v*/*v*) methanol and maintained at 4 °C in the dark overnight. After 12,000 rpm centrifugation for 10 min at 4 °C, the supernatants were collected, and the precipitates were further extracted in 0.5 mL of 80% methanol solution for 2 h at 4 °C and centrifuged again at 12,000 rpm for 10 min. Resultant supernatants were then combined, concentrated to dryness under a vacuum, discolored with an equal volume of petroleum ether three times, and filtered through a 0.22 μm syringe-driven filter. Extracted hormones were purified using a Waters Sep-pak C18 cartridge (Milford, MA, USA). The hormone analysis was performed on a Shimadzu LC 20A HPLC system (Kyoto, Japan) with absorbance of 254 nm in a UV detector. Results were expressed as ng/g FW for each endogenous hormone, and three replicates were carried out for each assay.

To detect the activity of superoxide dismutase (SOD), peroxidase (POD), and catalase (CAT), 1.0 g fresh samples were homogenized with 10 mL of ice-cold sodium phosphate buffer (50 mM, pH 7.0) in a chilled mortar. The homogenate was centrifuged at 10,000 rpm for 15 min at 4 °C. The resulting supernatant was used for the determination of the enzyme activity using the corresponding assay kits (Solarbio Science & Technology Co., Ltd., Beijing, China). The enzyme activity was expressed as U/g FW. Each treatment was repeated three times.

### 4.9. Statistical Analysis

Excel 2016 (Microsoft Corp., Redmond, WA, USA) and SPSS 23.0 (IBM Inc., New York, NY, USA) were used for experimental data processing. Before analysis, the contamination rate, survival rate, shoot induction frequency, and root induction frequency were transformed into the arcsine-square root. All data relating to surface disinfection and shoot induction were analyzed by one-way analysis of variance (ANOVA) with Duncan’s multiple range test, and statistically significant differences were determined at *p* < 0.05. Two-way ANOVA was performed to assess the effects of PGRs on shoot multiplication and in vitro rooting. The statistical significance of differences between the mean values of endogenous hormone content and enzyme activity was determined by Student’s *t*-test. The Pearson correlation coefficient method was used to reveal the relationships between the endogenous hormone content and antioxidant enzyme activity. The graphs were generated using Origin 2022 (OriginLab Inc., Northampton, MA, USA). 

## 5. Conclusions

The protocols of shoot induction, proliferation, rooting, acclimatization, and field planting were established from the nodal segments of the Jerusalem artichoke. A 4 min treatment with 0.1% HgCl_2_ was optimal for the surface disinfection of the stem nodes. The most efficient medium for shoot induction, shoot multiplication, and root induction was the MS medium containing 0.5 mg/L 6-BA, 0.5 mg/L 6-BA and 0.5 mg/L GA_3_, and 0.05 mg/L IBA and 0.05 mg/L NAA, respectively. Flow cytometry and SSR analysis confirmed the genetic stability of the regenerated plants. Furthermore, atypical shoots characterized by etiolated leaves and excessive calli were observed on the medium containing 0.5 mg/L 6-BA, which may lead to certain losses of plant material. The significantly higher levels of endogenous IAA and ABA, as well as the activity of CAT, SOD, and POD in the atypical shoots, could have been responsible for their aberrant growth and morphology. Overall, the established in vitro culture protocol provides a reliable approach to the clonal propagation of Jerusalem artichoke, and the possible physiological mechanisms underlying the formation of the atypical shoots might serve as a foundation for the further optimization of the in vitro propagation of this species.

## Figures and Tables

**Figure 1 plants-12-03789-f001:**
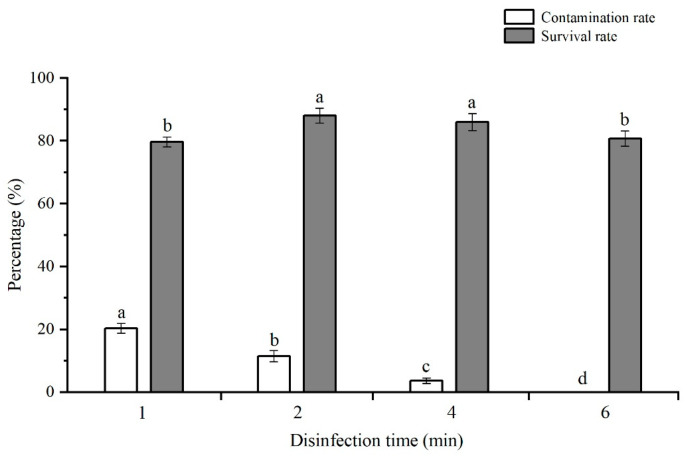
Effects of different dipping times of 0.1% (*w*/*v*) HgCl_2_ on the contamination rate and survival rate of the node explants of Jerusalem artichoke after 30 days of in vitro culture. The error bars represent the standard deviations (SD) of three replicates, and different letters above the error bars indicate significant differences according to Duncan’s multiple range test at *p* < 0.05.

**Figure 2 plants-12-03789-f002:**
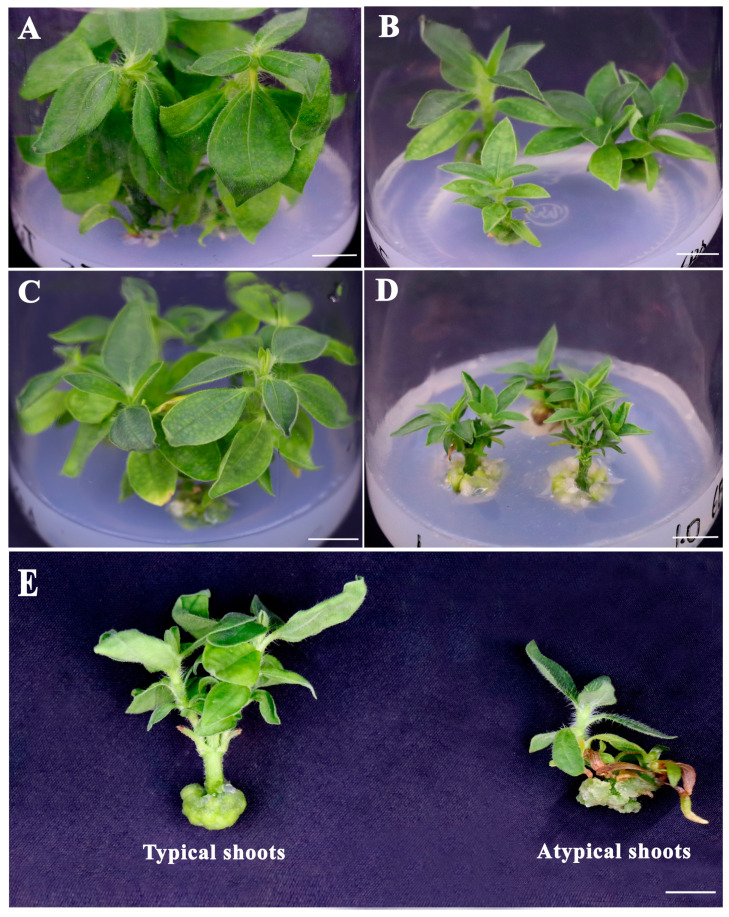
Shoot induction from the stem nodes of *Jerusalem artichoke* on MS basal media containing different concentrations of 6-BA after 30 days of culture. Regenerated shoots on the SI1 medium without 6-BA (**A**), the SI2 medium containing 0.25 mg/L 6-BA (**B**), the SI3 medium containing 0.5 mg/L 6-BA (**C**), and the SI4 medium containing 1.0 mg/L 6-BA (**D**)**,** respectively. (**E**) Typical and atypical shoots observed on the SI3 medium. Bars = 1 cm.

**Figure 3 plants-12-03789-f003:**
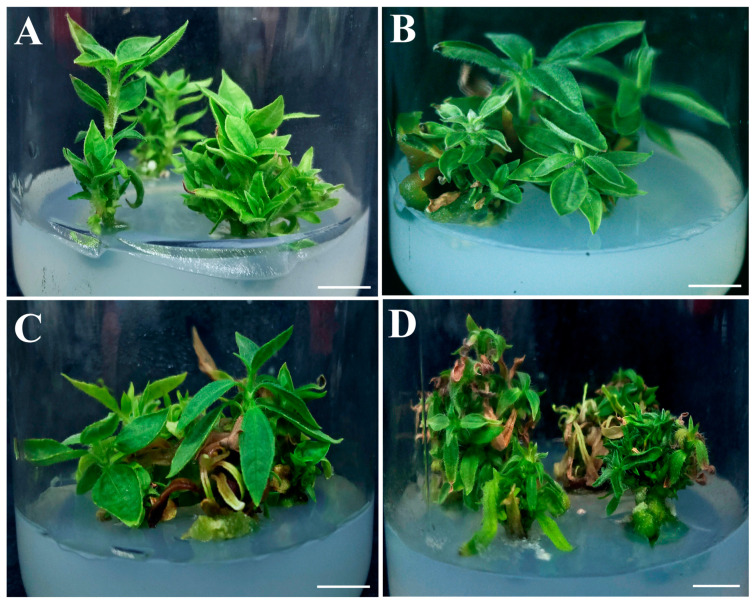
Shoot proliferation of Jerusalem artichoke on MS basal media supplemented with different concentrations of 6-BA and GA_3_ after 30 days of culture. The shoots regenerated on the SM1 medium containing 0.5 mg/L 6-BA and 0.1 mg/L GA_3_ (**A**), the SM2 medium containing 0.5 mg/L 6-BA and 0.5 mg/L GA_3_ (**B**), the SM3 medium containing 1.0 mg/L 6-BA and 0.1 mg/L GA_3_ (**C**), and the SM4 medium containing 1.0 mg/L 6-BA and 0.5 mg/L GA_3_ (**D**), respectively. Bars = 1 cm.

**Figure 4 plants-12-03789-f004:**
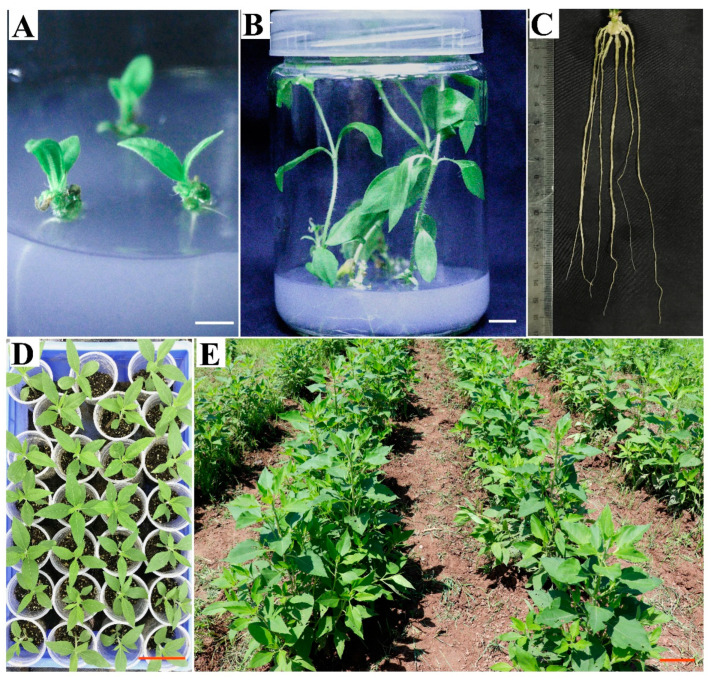
Root induction, plant acclimatization ex vitro, and plants growing in soil. (**A**) Induced roots after 12 days of culture on MS medium containing 0.05 mg/L IBA and 0.05 mg/L NAA (the RI2 medium). (**B**) In vitro regenerated plantlets cultured on the RI2 medium for 30 days. (**C**) Well-developed roots of the plantlets cultured on the RI2 medium for 30 days. (**D**) Surviving plants after 30 days of acclimatization. (**E**) Vigorously growing plants after 2 months of growth in the field. Bars = 1 cm for (**A**,**B**), and 10 cm for (**D**,**E**).

**Figure 5 plants-12-03789-f005:**
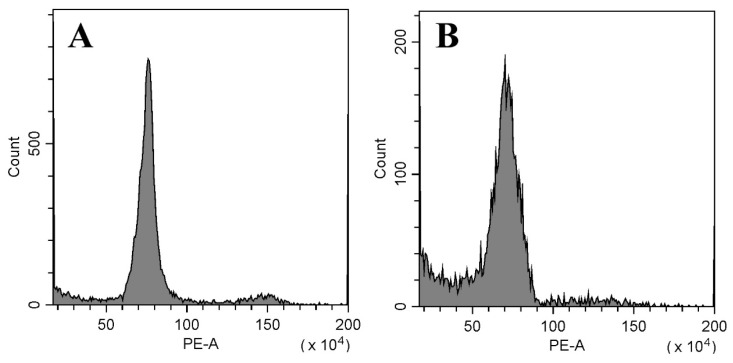
Flow cytometry analysis of the Jerusalem artichoke donor plants (**A**) and the in vitro regenerated plantlets (**B**).

**Figure 6 plants-12-03789-f006:**
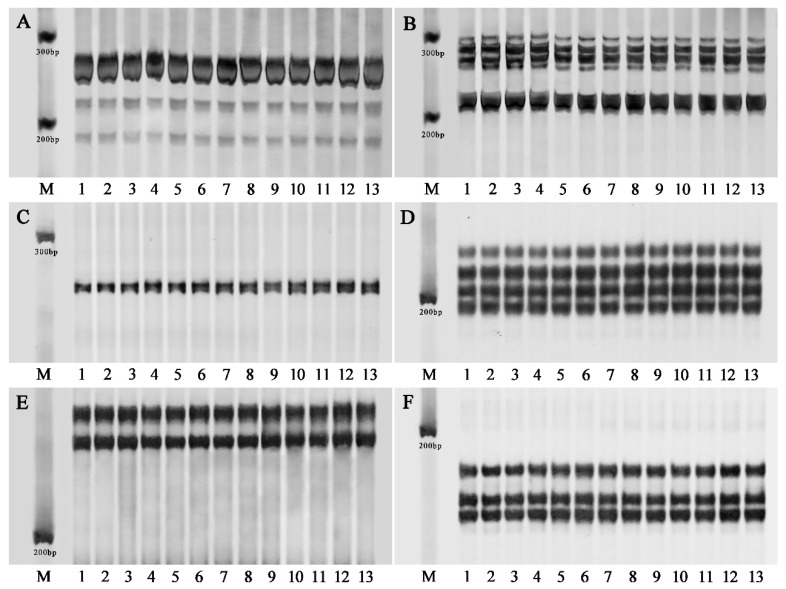
SSR profiles of the donor plant and 12 randomly selected plantlets regenerated in vitro from stem nodes of Jerusalem artichoke. Banding patterns amplified by the primer pairs JS5 (**A**), JS8 (**B**), JS13 (**C**), JS18 (**D**)**,** JS19 (**E**), and JS20 (**F**), respectively. M: DL500 DNA marker (Takara Biotechnology Co., Ltd., Dalian, China); 1: the donor plant; 2–13: the randomly selected in vitro regenerated plantlets.

**Figure 7 plants-12-03789-f007:**
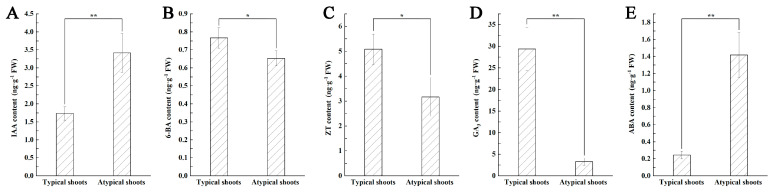
Comparisons of the content of endogenous hormones in the typical and atypical shoots obtained on the SI3 medium. (**A**) IAA. (**B**) 6-BA. (**C**) ZT. (**D**) GA_3_. (**E**) ABA. The error bars represent the standard deviations (SD) of three replicates, and * and ** indicate significant differences between the two types of shoots estimated by Student’s *t*-test at *p* < 0.05 and 0.01, respectively.

**Figure 8 plants-12-03789-f008:**
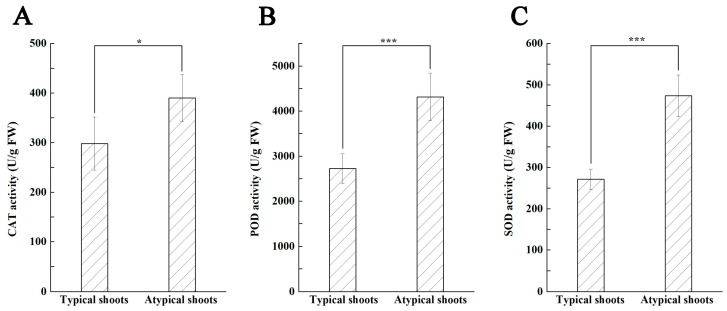
Comparisons of the antioxidant enzyme activity in the typical and atypical shoots obtained on the SI3 medium. (**A**) CAT. (**B**) POD. (**C**) SOD. The error bars represent the standard deviations (SD) of three replicates, and * and *** indicate significant differences between the two types of shoots estimated by Student’s *t*-test at *p* < 0.05 and 0.001, respectively.

**Figure 9 plants-12-03789-f009:**
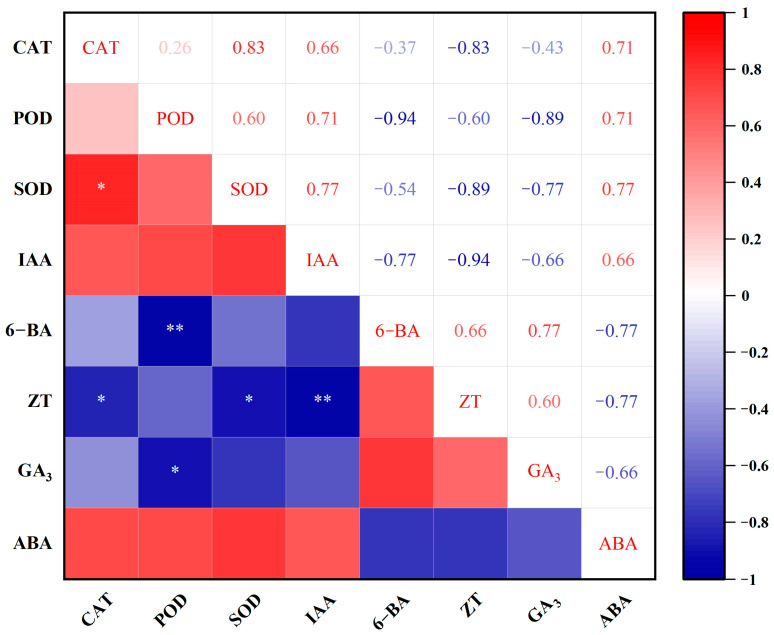
Heat map of correlation analysis between the endogenous hormone levels and antioxidant enzyme activity in the typical and atypical shoots. * and ** indicate significant correlation at 5% and 1% level, respectively.

**Table 1 plants-12-03789-t001:** Effects of different concentrations of 6-BA on shoot induction and shoot morphological characteristics of Jerusalem artichoke.

Medium	6-BA (mg/L)	Shoot Induction Frequency (%)	No. of Regenerated Shoots per Explant	Node Number per Shoot	Shoot Length (cm)	No. of Leaves per Shoot	Fresh Weight of the Calli Formed at the Explant Base (mg)	Frequency of Atypical Shoots (%)
SI1	0	94.4 ± 8.2 a	1.1 ± 0.1 c	5.7 ± 1.1 a	4.6 ± 0.8 a	11.8 ± 1.1 ab	36.6 ± 6.3 c	0.0 ± 0.0 d
SI2	0.25	90.2 ± 2.9 ab	1.5 ± 0.4 c	3.9 ± 0.7 b	2.3 ± 0.5 b	9.3 ± 1.0 c	468.1 ± 57.1 b	8.3 ± 0.9 c
SI3	0.5	87.2 ±1.7 b	2.5 ± 0.7 b	6.5 ± 1.1 a	3.9 ± 0.8 a	13.4 ± 2.4 a	502.3 ± 41.3 b	17.3 ± 4.8 b
SI4	1.0	85.2 ± 3.0 b	3.6 ± 0.8 a	4.3 ± 0.8 b	1.8 ± 0.3 b	10.3 ± 1.3 bc	728.1 ± 167.4 a	28.0 ± 6.2 a

Means ± standard deviation in a column followed by different letters indicate significant difference at 5% level according to Duncan’s multiple range test.

**Table 2 plants-12-03789-t002:** Effects of different concentrations of 6-BA and GA_3_ on shoot multiplication of Jerusalem artichoke after 30 days of in vitro culture.

Medium	PGRs (mg/L)		No. of Shoots per Explant		Shoot Length (cm)		No. of Leaves per Shoot	
	6-BA	GA_3_						
SM1	0.5	0.1	2.9 ± 0.8 b		3.1 ± 0.3 a		5.7 ± 1.4 ab	
SM2	0.5	0.5	3.2 ± 0.7 ab		3.3 ± 0.4 a		6.1 ± 1.5 a	
SM3	1.0	0.1	3.0 ± 0.8 b		2.5 ± 0.6 b		4.8 ± 1.1 bc	
SM4	1.0	0.5	3.7 ± 1.1 a		1.7 ± 0.2 c		4.3 ± 1.2 c	
ANOVA summary table								
Source			F value	*p* value	F value	*p* value	F value	*p* value
6-BA			1.568	ns	106.458	***	13.476	***
GA_3_			5.961	*	8.634	**	0.005	ns
6-BA × GA_3_			0.581	ns	20.907	***	1.432	ns

Means ± standard deviation within a column followed by the same letter does not differ significantly at 5% level according to Duncan’s multiple range test. ns: not significant; *, **, and ***: Level of significance at 0.05, 0.01, and 0.001, respectively.

**Table 3 plants-12-03789-t003:** Effects of different concentrations of IBA and NAA on rooting of Jerusalem artichoke after 30 days in culture.

Medium	PGRs (mg/L)		Root Induction Frequency (%)		No. of Adventitious Roots per Plantlet		Average Root Length (cm)	
	IBA	NAA						
RI1	0.05	0	87.5 ± 12.5 a		3.3 ± 0.2 c		12.2 ± 1.7 a	
RI2	0.05	0.05	95.8 ± 7.2 a		4.7 ± 0.8 a		14.7 ± 2.0 a	
RI3	0.1	0	85.0 ± 4.3 a		3.9 ± 0. 5 bc		13.3 ± 1.5 a	
RI4	0.1	0.05	95.2 ± 8.3 a		4.4 ± 0.2 ab		12.4 ± 0.4 a	
ANOVA summary table								
Source			F value	*p* value	F value	*p* value	F value	*p* value
IBA			0.097	ns	0.205	ns	0.485	ns
NAA			3.506	ns	12.023	**	0.820	ns
IBA × NAA			0.037	ns	2.750	ns	3.978	ns

Means ± standard deviation within a column followed by the same letter does not differ significantly at 5% level according to Duncan’s multiple range test. ns: not significant; **: Level of significance at 0.01.

## Data Availability

The data are available upon request from the corresponding authors.

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
