# Peer review of "Plant Regeneration via Organogenesis in Jerusalem Artichokes and Comparative Analysis of Endogenous Hormones and Antioxidant Enzymes in Typical and Atypical Shoots"

_plants, 2023, doi:10.3390/plants12223789_

Round 1
Reviewer 1 Report
Comments and Suggestions for Authors
see PDF

Comments on the Quality of English LanguageModerate editing of English language required
Reviewer 2 Report
Comments and Suggestions for Authors
The detailed comments and questions are given in the file enclosed.
Additionally:
1. 12 explants per replication is very little. It might be understandable for the initiation stage if it’s difficult to obtain primary explants (however it does not seem difficult in the case of Jerusalem artichoke), however for further stages the Authors could have prepared more plant material.
2. Figures 1, 5, 6, 7, 8 and 9 are of a very poor quality – hardly readable.
3. The supplementary Figure should be included in the main manuscript.

Comments on the Quality of English LanguageEnglish mostly fine, a few sentences are too long and complicated, a few minor errors.
Round 2
Reviewer 1 Report
Comments and Suggestions for Authors
dear author
I have read the new version very carefully. The new version is much improved compared to the first one. However, I still had comments which I commented on the new version of the MS. Please relate to the new comments in the attached file.
After fully relating to the new comments the MS can be published
Please when you write the cover letter, mark the line number for each comment and not by serial number

Author Response
Dear reviewer,
Thank you very much for your careful review of our manuscript. The attachment is our reponses to your comments.
